# Fourier Spectrum Discrepancies in Deep Network Generated Images

**Tarik Dzanic**
Department of Ocean Engineering
Texas A&M University
College Station, TX 77843
tdzanic@tamu.edu

**Karan Shah**
Department of Computational Science and Engineering
Georgia Institute of Technology
Atlanta, GA 30332
shah@gatech.edu

**Freddie D. Witherden**
Department of Ocean Engineering
Texas A&M University
College Station, TX 77843
fdw@tamu.edu

## Abstract

Advancements in deep generative models such as generative adversarial networks and variational autoencoders have resulted in the ability to generate realistic images that are visually indistinguishable from real images, which raises concerns about their potential malicious usage. In this paper, we present an analysis of the high-frequency Fourier modes of real and deep network generated images and show that deep network generated images share an observable, systematic shortcoming in replicating the attributes of these high-frequency modes. Using this, we propose a detection method based on the frequency spectrum of the images which is able to achieve an accuracy of up to 99.2% in classifying real and deep network generated images from various GAN and VAE architectures on a dataset of 5000 images with as few as 8 training examples. Furthermore, we show the impact of image transformations such as compression, cropping, and resolution reduction on the classification accuracy and suggest a method for modifying the high-frequency attributes of deep network generated images to mimic real images.

## 1 Introduction

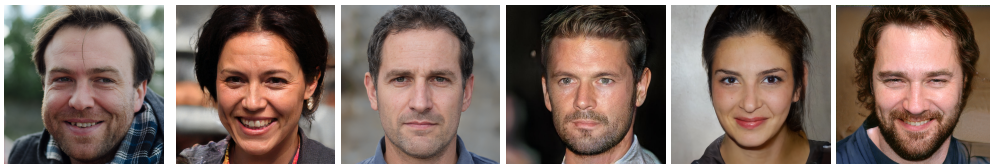

Figure 1: Left to right: real, StyleGAN [1], StyleGAN2 [2], PGGAN [3], VQ-VAE2 [4], and ALAE [5] generated images.

In recent years, advances in deep generative models for image synthesis have caused widespread concern regarding their potential malicious uses. Current state-of-the-art models can generate hyper-realistic images that are visually indistinguishable from real images, as shown in Fig. 1. These models can be used for unethical purposes, such as misinformation campaigns, fabricating evidence,

or attacking biometric security systems, and as a result, recent research efforts have been focused on developing methods for detecting such images [6, 7]. Various detection strategies have been employed, from traditional image forensics methods such as analyzing encoding or acquisition fingerprints to, more recently, machine learning based approaches such as statistical modeling for disparities in color components, textures, or features [8, 9, 10, 11, 12]. Boulkenafet *et al.* developed a method for face spoofing detection by analyzing the statistical distributions of the image spectra in other color spaces such as HSV and YCbCr [9]. Li *et al.* expanded upon this method to detect images generated by generative adversarial networks [10]. Marra *et al.* showed that generative adversarial networks leave unique artificial fingerprints in the noise of their generated images that are dependent on the network architecture which can be used to detect if an image was generated by a particular network [11]. In concurrent work, Wang *et al.* trained a deep neural network for classifying deep network generated images and showed that a classifier trained on one generative model can reasonably generalize to other models as well [13]. Although these methods have shown promise in terms of accuracy with minimal user input, they require large amounts of data to train, something which may not be feasible in many applications. As such, there is a need for detection methodologies that can obtain similar levels of accuracy with minimal training data that can generalize to unknown generative models.

In this work, we explore the high-frequency characteristics of real and deep network generated images and propose a simple yet robust detection method for deep network generated images based on these characteristics. We compare real images to images generated by various generative models – generative adversarial networks [14] such as StyleGAN [1], StyleGAN2 [2], and PGGAN [3] as well as variational and hybrid autoencoders [15] such as VQ-VAE2 [4] and ALAE [5]. Based on a reduced-order model of the high-frequency spectrum of the images, we show that the relative magnitude and the decay rate of the high-frequency spectrum is clearly distinguishable between real and deep network generated images, indicating that the low-level attributes of images generated by generative models display different properties. Furthermore, we show that at higher resolutions, these differences are more easily observed, and we consider the impact of common image transformation methods such as compression and cropping which can significantly affect the high-frequency spectra, causing difficulties in detecting deep network generated images. We also show the results of binary classification experiments based on these low-level attributes of images at different resolutions and compression levels from datasets such as Flickr-Faces-HQ (FFHQ) and the aforementioned generative models. Finally, we present a method for modifying the spectra of deep network generated images to mimic the high-frequency characteristics of real images as a way of deceiving such classifiers.

## 2 Methodology

### 2.1 Fourier spectrum analysis

To analyze the characteristics of real and deep network generated images in the frequency domain, a Fourier transform is required. For a discrete two-dimensional signal $f(p, q)$ representing individual color channels of an image of size $m \times n$, the discrete Fourier transform $\mathcal{F}(k_x, k_y)$ is defined as

$$\mathcal{F}(k_x, k_y) = \frac{1}{mn} \sum_{p=0}^{m-1} \sum_{q=0}^{n-1} f(p, q) e^{-i2\pi(k_x p/m + k_y q/n)},$$ (1)

which is of the same dimension as the input signal. To construct a scale and rotation invariant threshold for the highest frequencies, a transform in wavenumber space can be performed from Cartesian coordinates $k_x, k_y$ to normalized polar coordinates $k_r \in [0, 1]$ and $\theta \in [0, 2\pi)$.

$$\mathcal{F}(k_r, \theta) = \mathcal{F}(k_x, k_y) \quad : \quad k_r = \sqrt{\frac{k_x^2 + k_y^2}{\frac{1}{4}(m^2 + n^2)}}, \ \theta = \text{atan2}\,(k_y, k_x).$$ (2)

Furthermore, the dimensionality can be reduced without significant loss in information by azimuthally averaging the magnitude of the Fourier coefficients to obtain the reduced spectrum $c(k_r)$, a quantitative representation of the strength of the signal with respect to the radial wavenumber $k_r$

$$c(k_r) = \frac{1}{2\pi} \int_0^{2\pi} \left| \mathcal{F}(k_r, \theta) \right| d\theta.$$ (3)

In practice, this averaging is approximated with binning along the radial direction to smooth the large fluctuations in the Fourier spectrum at high frequencies. Although a classifier can be trained on the reduced spectrum, a simpler and more robust classifier can be built by fitting a decay function to the reduced spectrum and classifying using the parameters of this function. As the spectra of natural images tend to behave as a power law [16], performing classification on a power law decay function is considered in this work, modeled as

$$c(k_r) \approx b_1 \left( \frac{k_r}{k_T} \right)^{b_2}, \qquad k_r \in [k_T, 1], \tag{4}$$

where the parameter $k_T$ denotes a threshold wavenumber above which the fitting is performed. With this approximate form, the high-frequency spectrum is represented by two independent parameters: $b_1$, which represents the magnitude of the high-frequency content, and $b_2$, which represents the decay rate of the high-frequency spectrum. These parameters, along with the reduced spectra, are used to highlight differences in the high-frequency characteristics of real and deep network generated images.

## 2.2 Image transformations

In order to minimize storage and bandwidth requirements, deep network generated images are typically resized, cropped, and/or compressed, procedures which can change the characteristics of the image spectra. The image resolution dictates the maximum frequency in the frequency domain, and higher resolutions yield more information at the highest frequencies. Compression may particularly affect the high-frequency spectrum of an image since the high-frequency components of an image correspond to the small scale features whereas the low-frequency components correspond to the large features. Therefore, compression algorithms generally tolerate losses in the high-frequency components as they have less impact on the way an image is seen compared to the low-frequency components [17]. Common compression methods such as quantization and subsampling can spuriously introduce or reduce high-frequency content, respectively [18].

Images with varying resolutions – both native and cropped – and compression levels were analyzed in this work. Datasets of different resolutions were used in the experiments, and compression levels were varied using lossy JPEG compression with Python Imaging Library (Pillow). A quality metric is given to indicate the amount of compression. For the 100% quality images, the original provided images were used, consisting of either lossless PNG or 100% quality JPEG images. Although the latter is not considered lossless, negligible differences in the reduced spectra between the two image formats were seen, and therefore both of these are referred to as uncompressed in this work. Two additional compression levels were chosen corresponding to high quality (95%) and medium quality (85%) compression. The latter was chosen qualitatively based on the visually noticeable presence of compression artifacts while the former was chosen as it is a default setting in many applications.

## 2.3 Classification

### 2.3.1 Datasets

A binary classification task was performed between various real and deep network generated images. Image samples were taken from datasets of real images and images generated by StyleGAN, Style-GAN2, PGGAN, VQ-VAE2 and ALAE architectures at compression qualities of 100%, 95%, and 85%. These datasets, shown in Table 1, are denoted by $\mathcal{R}, \mathcal{G}, \mathcal{S}, \mathcal{P}, \mathcal{V}$, and $\mathcal{A}$, respectively, with the subscript denoting the resolution. Additional datasets, denoted with the subscript 768, were created by taking the native $1024^2$ resolution datasets and cropping them to a resolution of $768^2$.

For the majority of the datasets, 10% of the images were used for training while the remaining 90% were used for testing to highlight the relatively low number of training examples required for classification. For the high-resolution VQ-VAE2 datasets ($\mathcal{V}_{1024}/\mathcal{V}_{768}$), only a small number of high-resolution images were presented in the work by Razavi *et al.* [4], and therefore only 8 images were available for training and 9 for testing. For the low-resolution VQ-VAE2 dataset ($\mathcal{V}_{256}$), a larger amount of low-resolution images were provided and 100 of the 364 images were used for training while the remaining were used for testing. In both cases, these images were duplicated to match the size of the other datasets to give equal weight to the training and testing metrics.

### 2.3.2 Classifier

To emphasize the postulate that the low-level properties of real and deep network generated images are fundamentally different, the classification was performed using only "simple" classifiers. A k-nearest neighbors (KNN) classifier with $k = 5$ was used for classification between real and deep network generated images with respect to the decay parameters $(b_1, b_2)$ of the grayscale component of the images. Since the data was easily separable in many cases, negligible differences in classification accuracy were obtained with other KNN hyperparameter choices and with different classifiers such as a support vector machine with a variety of kernel choices. As the classification was performed with respect to only two parameters, minimal training data was required and the computational cost of training and classifying was insignificant.

Classification accuracy was determined by the ability of the classifier to predict if an image was real or fake, and no weight was placed on discerning between the architecture that generated the images. Overall classification accuracy was calculated using the real image datasets and the datasets from each generative model. Individual classification accuracies for each generative model were separately calculated from a subset of training and testing data using only real images and images from the respective model. The pipeline for the classification task was as follows:

1. Perform the discrete Fourier transform of the image and normalize by the DC gain.

2. Transform from Cartesian coordinates to normalized polar coordinates in the frequency domain.

3. Bin the magnitudes of the Fourier coefficients along the radial direction and average azimuthally to obtain the reduced spectrum.

4. Fit the decay parameters $b_1, b_2$ to the reduced spectrum above a threshold wavenumber $k_T$.

5. Train/apply the binary classifier to the decay parameters of the image to predict if the image is real or fake.

## 3 Experiments and results

In this section, the reduced spectrum for the images from the datasets in Table 1 is shown as well as the effects of resolution, cropping, and compression on the spectra. Additionally, experimental results for the classification task between real and deep network generated images at different resolutions, cropping levels, and compression qualities are presented.

### 3.1 Reduced spectrum

A comparison of the reduced spectrum statistics of the grayscale-converted $1024^2$ pixel images from the datasets in Table 1 is shown in Fig. 2, normalized by the spectrum at a threshold wavenumber

Table 1: Experimental datasets

| Dataset | Origin | Dataset Type | Resolution | Compression Quality | Training Samples | Testing Samples |
|---|---|---|---|---|---|---|
| $\mathcal{R}_{1024}$ | FFHQ | Faces | $1024^2$ | [100, 95, 85] | 100 | 900 |
| $\mathcal{G}_{1024}$ | Karras *et al.* [1] | Faces | $1024^2$ | [100, 95, 85] | 100 | 900 |
| $\mathcal{S}_{1024}$ | Karras *et al.* [2] | Faces | $1024^2$ | [100, 95, 85] | 100 | 900 |
| $\mathcal{P}_{1024}$ | Karras *et al.* [3] | Faces | $1024^2$ | [100, 95, 85] | 100 | 900 |
| $\mathcal{V}_{1024}$ | Razavi *et al.* [4] | Faces | $1024^2$ | [100, 95, 85] | 8 | 9 |
| $\mathcal{A}_{1024}$ | Pidhorskyi *et al.* [5] | Faces | $1024^2$ | [100, 95, 85] | 100 | 900 |
| $\mathcal{R}_{256}$ | Zhang *et al.* [19] | Cats | $256^2$ | [100, 95, 85] | 100 | 900 |
| $\mathcal{G}_{256}$ | Karras *et al.* [1] | Cats | $256^2$ | [100, 95, 85] | 100 | 900 |
| $\mathcal{S}_{256}$ | Karras *et al.* [2] | Cats | $256^2$ | [100, 95, 85] | 100 | 900 |
| $\mathcal{P}_{256}$ | Karras *et al.* [3] | Cats | $256^2$ | [100, 95, 85] | 100 | 900 |
| $\mathcal{V}_{256}$ | Razavi *et al.*[4] | Animals | $256^2$ | [100, 95, 85] | 100 | 264 |
| $\mathcal{A}_{256}$ | Pidhorskyi *et al.* [5] | Faces | $256^2$ | [100, 95, 85] | 100 | 900 |

$k_T = 0.75$. At the threshold wavenumber, the real images show a decay initially proportional to approximately $k_r^{-4}$ before leveling off near the end of the spectrum. In contrast, the deep network generated images – with an exception of images generated by StyleGAN2 – do not show such decay, exhibiting decay exponents of less than 1. As the threshold wavenumber was increased, the StyleGAN2 images behaved similarly to the other deep network generated images. Similar results were observed with the spectra of the individual color channels as with the grayscale-converted images.

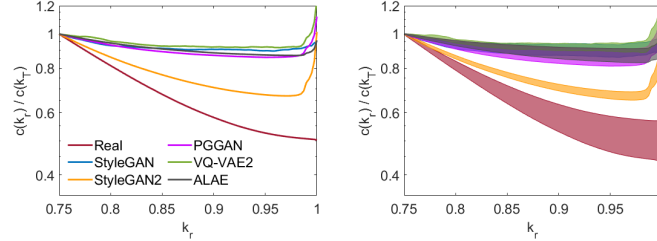

Figure 2: Normalized reduced spectra: mean (left) and $\pm 1$ standard deviation (right).

### 3.1.1 Resolution and cropping

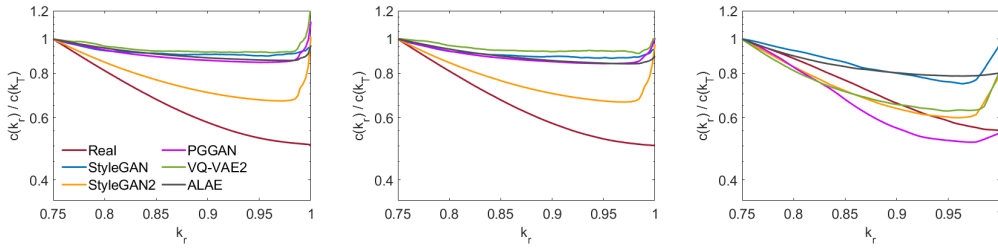

Figure 3: Mean normalized reduced spectra: $1024^2$ (left), cropped $768^2$ (middle), and $256^2$ (right).

The reduced spectrum was computed for the cropped $768^2$ and $256^2$ pixel images sampled from the datasets in Table 1. These spectra were compared to the $1024^2$ pixel image spectra from Fig. 2. In comparison to the $1024^2$ pixel images, the $768^2$ pixel image spectra behaved almost identically, as shown in Fig. 3. However, the $256^2$ pixel image spectra behaved noticeably different, with the deep network generated image spectra exhibiting lower decay rates whereas the real image spectra were qualitatively similar to the higher resolution images. As the resolution was lowered, it became more difficult to distinguish between the real and deep network generated image spectra as the maximum frequency was reduced. However, the same observations as with the higher resolution images can be drawn with the lower resolution images if the threshold wavenumber was increased as the tail of the deep network generated image spectra began to flatten at the highest wavenumbers.

### 3.1.2 Compression

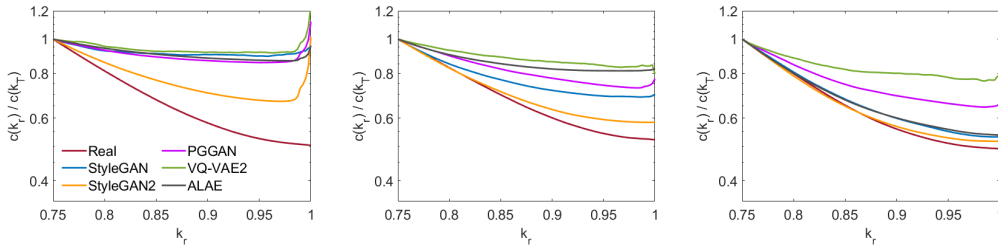

Figure 4: Mean normalized reduced spectra: 100% (left), 95% (middle), and 85% quality (right).

The effects of image compression on the reduced spectrum of the $1024^2$ pixel image datasets in Table 1 is shown in Fig. 4 for compression qualities of 100%, 95%, and 85%. Even at a small compression ratio (95%), the high-frequency reduced spectra of the deep network generated images were significantly modified, and their decay rate converged to the decay rate of the real images. At 85% compression, the StyleGAN, StyleGAN2, and ALAE image spectra were essentially indistinguishable from the real image spectra, with only slightly lower decay rates than the compressed real images. In contrast, the reduced spectra of the VQ-VAE2 and PGGAN images were less affected by the compression, with VQ-VAE2 images showing clearly distinguishable spectra even for compression qualities as low as 60%. The relative effects of compression on the decay rates of the high-frequency spectra can be directly attributed to the amount of high-frequency content. Lossy compression methods, whose effects are proportional to the frequency and effectively modify the decay rate, would have negligible impact if there was very little high-frequency content.

These observations indicate that compression, even in small amounts, acts to homogenize the spectral content of images generated by certain architectures. In a model-unaware scenario where the classifier does not know if the images are compressed, it would not be able to robustly distinguish between uncompressed real images, compressed real images, and compressed StyleGAN, StyleGAN2, and ALAE images, but would be able to easily distinguish their uncompressed counterparts. However, VQ-VAE2 and certain PGGAN images would remain easily distinguishable regardless of compression as their spectra's decay rates are less affected, and a different method for mimicking the spectrum of real images is needed.

## 3.2 Classification

The results of the KNN classifier for image resolutions of $1024^2$, $768^2$ (cropped), and $256^2$ with compression qualities of 100% (uncompressed), 95%, and 85% are shown in Table 2. When classifying uncompressed $1024^2$ images (experiment A), the classifier was able to obtain a 99.2% accuracy across all image types with a minimum and maximum accuracy of 97.4% (PGGAN) and 99.9% (StyleGAN), respectively, when classifying images generated by a single architecture.

For the uncompressed $1024^2$ images (experiment A), the distribution of the data along the $b_1 - b_2$ axes displayed distinct clusters corresponding to the various image types, as shown in Fig. 5a. Real images exhibited a range of high-frequency content ($b_1$) with notably high decay rates ($-b_2$). All deep network generated images had significantly smaller decay rates than the real images, but the high-frequency content of images from each generative model varied with VQ-VAE2 and ALAE producing the lowest and highest amount of high-frequency content, respectively. In contrast to the reduced spectrum statistics shown in Fig. 2 where the StyleGAN2 images were most similar to the real images, the PGGAN images were instead more likely to be misclassified. This can be attributed to the increased high-frequency content of the StyleGAN2 images which allowed the classifier to distinguish the images from real images even though the decay rates were more similar. The lower level of high-frequency content in the PGGAN images made the images more similar to certain real images with low decay rates which caused instances of misclassification.

As the images were compressed (experiments B-C), the distribution of the decay rates of the real and deep network generated images converged and many of the clusters were indistinguishable at 85% compression quality, as shown in Fig. 5c, where the overall classification accuracy was

Table 2: Classification experiments and results

| Experiment | Resolution | Compression Quality | Overall Class. Acc. | StyleGAN Class. Acc. | StyleGAN2 Class. Acc. | PGGAN Class. Acc. | VQ-VAE2 Class. Acc. | ALAE Class. Acc. |
|---|---|---|---|---|---|---|---|---|
| A | $1024^2$ | 100 | **99.2%** | 99.9% | 99.5% | 97.4% | 99.8% | 99.8% |
| B | $1024^2$ | 95 | **94.4%** | 99.2% | 88.5% | 88.5% | 100 % | 99.7% |
| C | $1024^2$ | 85 | **83.9%** | 78.9% | 65.9% | 78.7% | 99.6% | 87.4% |
| D | $768^2$ | 100 | **98.5%** | 100 % | 99.1% | 95.9% | 99.9% | 99.9% |
| E | $768^2$ | 95 | **93.0%** | 97.9% | 85.4% | 87.3% | 100 % | 99.5% |
| F | $768^2$ | 85 | **84.6%** | 77.1% | 68.6% | 79.3% | 99.6% | 85.7% |
| G | $256^2$ | 100 | **88.8%** | 85.0% | 87.4% | 69.0% | 92.0% | 90.7% |
| H | $256^2$ | 95 | **88.1%** | 81.7% | 83.4% | 68.2% | 92.2% | 87.7% |
| I | $256^2$ | 85 | **87.4%** | 67.8% | 79.3% | 64.8% | 87.7% | 80.6% |

reduced to 83.9%. Nearly identical observations were drawn from the cropped $768^2$ pixel images (experiments D-F, not shown), and the overall and individual classification accuracies were generally within 1-2% of their native resolution counterparts. Furthermore, as the decay fitting method in Eq. 4 is independent of the resolution, the classifier trained on $1024^2$ pixel images was able to maintain a similar classification accuracy when classifying $768^2$ pixel images, demonstrating the robustness of the method.

Due to their small high-frequency content, the effects of compression on the VQ-VAE2 images were minimal, and as such, the VQ-VAE2 images were correctly classified regardless of compression quality for both $1024^2$ and $768^2$ pixel images. The reason for the notable decrease of high-frequency content in VQ-VAE2 images is not immediately evident. It is hypothesized that VAEs tend to distribute probability mass diffusely over the data space, and thus their generated images tend to be blurry [20, 21, 22]. Although this is not visually noticeable, it is noticeable in the frequency domain as the high-frequency content associated with sharp edges is dramatically reduced.

When the classifier was tested on the $256^2$ images (experiments G-I), the classification accuracy was significantly lower. Even for the uncompressed images, the data was not as clearly separable as with the $1024^2$ images, and the classifier was only able to obtain an 88.8% overall classification accuracy. However, the overall distribution trend along the $b_1 - b_2$ axes of the various image types was similar at low and high resolutions as shown in Fig. 5a and Fig. 6a. Compression did not have as large of an effect on the $256^2$ images, and thus the classifier performed only slightly worse (87.4%) at 85% compression quality than on the uncompressed images. In contrast to the high-resolution experiments, the effects of compression on the the decay rates were minimal.

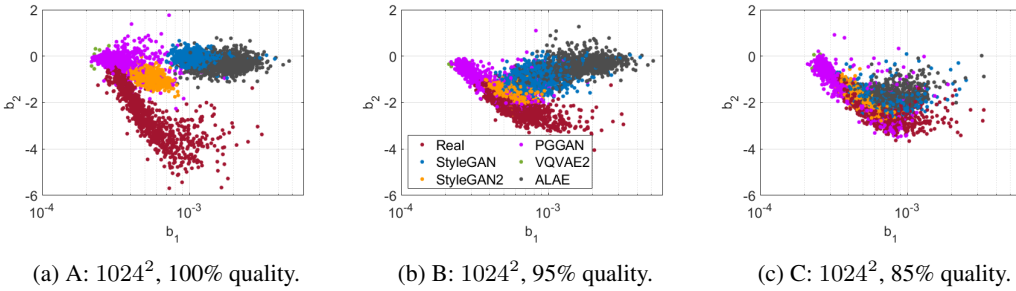

(a) A: $1024^2$, 100% quality.    (b) B: $1024^2$, 95% quality.    (c) C: $1024^2$, 85% quality.

Figure 5: Experiments A-C at $1024^2$ resolution.

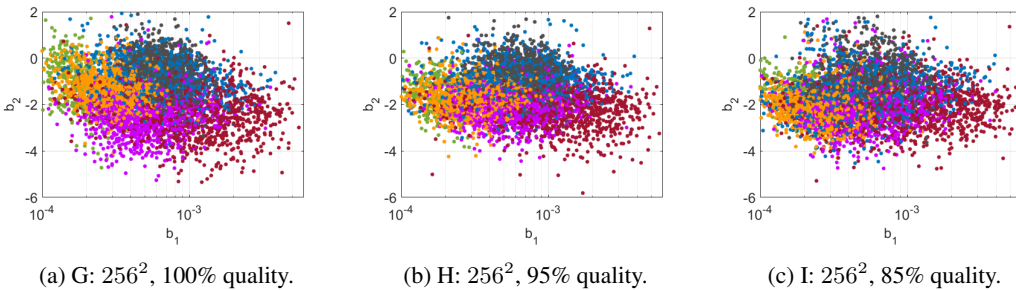

(a) G: $256^2$, 100% quality.    (b) H: $256^2$, 95% quality.    (c) I: $256^2$, 85% quality.

Figure 6: Experiments G-I at $256^2$ resolution.

## 3.3  Discussion

The cause of the disparities in the decay rates of the high-frequency content of deep network generated images is of particular interest as this issue is evident in each one of the investigated algorithms. One might believe that regularization has an effect on the high-frequency attributes as deep generative networks are not incentivised to learn the high-frequency components (i.e. noise) of their input data to discourage overfitting. However, we continued training the ALAE model with and without regularization, and we observed negligible differences in the decay rates of images generated by the model. In the work of Durall *et al.*, discrepancies in the high-frequency content of deep network

generated images were attributed to the effects of up-convolution [23]. Although this is shown to have an impact on the spectral composition of the images, it did not necessarily affect the decay rates (see Fig. 5). Instead, these discrepancies are most likely explained by the work of Khayatkhoei and Elgammal which presents an analysis on the spectral bias of convolution layers in a deep generative network [24]. They showed that linear dependencies exist in a convolution layer's filter spectrum which results in correlations between frequency components that are generally more pronounced at high frequencies. Consequently, these correlations can cause the flatlining of the frequency spectrum at high frequencies as seen in Fig. 2.

## 4 Spectrum synthesis

As the results in the classification experiments show, classifiers based on the high-frequency characteristics of images can easily distinguish between real and deep network generated images in many cases. In some scenarios, compression can effectively disguise these deep network generated images to the classifier, whereas in other scenarios, it has little effect. However, in scenarios where compression is a viable spoofing tool, the amount of compression required generally introduces noticeable visual artifacts.

To robustly disguise deep network generated images to classifiers based on high-frequency spectrum characteristics, a post-processing method for modifying the spectra of the deep network generated images to behave as real image spectra is proposed. Given the spectrum of a real image as a target, the high-frequency components of a deep network generated image were scaled to match the real image to produce a spoofed spectrum $\bar{\mathcal{F}}(k_r, \theta)$, which was then transformed back to a spoofed image. The scaling factor was defined as the ratio of the fitted decay functions of the target (real) and source (deep network generated) images.

$$\bar{\mathcal{F}}(k_r, \theta) = \mathcal{F}(k_r, \theta) \left[ \left(1 - \phi(k_r)\right) + \phi(k_r) \frac{b_{1,t}}{b_{1,s}} \left(\frac{k_r}{k_T}\right)^{(b_{2,t} - b_{2,s})} \right] \quad (5)$$

A smooth hyperbolic tangent blending function $\phi(k_r)$ was used to leave the low-frequency components of the image unaffected without introducing visual artifacts.

$$\phi(k_r) = \frac{1}{2}(\tanh(k_r - k_T) + 1) \quad (6)$$

The effects of the spectrum synthesis method on the reduced spectrum of the example VQ-VAE2 image are shown in Fig. 7. Using the real image in Fig. 1 as the target, the spoofed spectrum matched the real image spectrum very closely in both decay and magnitude and was visually indistinguishable from the original image. When compared to its compressed counterpart, the spoofed image was of noticeably higher quality since the spectrum synthesis method did not introduce compression artifacts, as shown by the pixel difference plot in Fig. 7b. The spoofed image fell well within the classification boundary for real images and effectively disguised the image to the classifier whereas compression could not. Similar results were obtained with other generative models.

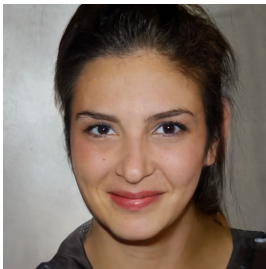
(a) Spoofed image

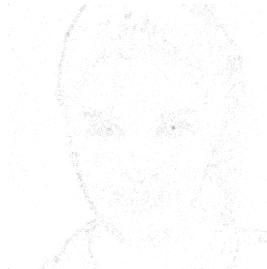
(b) Original/spoofed pixel difference (scaled ×100)

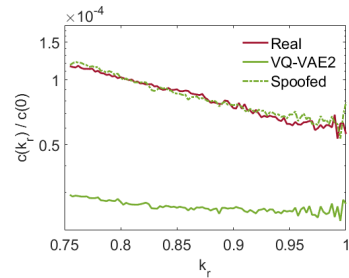
(c) Reduced spectra normalized by DC gain

Figure 7: Spectrum synthesis method.

# 5 Conclusion

In this work, we presented an analysis of the high-frequency modes of real images and images generated by various generative models. We showed that the Fourier modes of deep network generated images at the highest frequencies did not decay as seen in real images but instead stayed approximately constant. By modeling the decay of the Fourier spectrum at high frequencies, we observed that the high-frequency spectra of real and deep network generated images had distinct characteristics: real images showed large decay rates and a range of magnitudes, whereas deep network generated images showed small decay rates and the magnitude varied depending on the generative model. These differences were more noticeable at higher resolutions, but lossy image compression algorithms modified the high-frequency spectra and reduced those differences. When highly compressed, images generated by certain architectures were indistinguishable from real images in the frequency domain, but images generated by other architectures like VAEs were not affected due to their low levels of high-frequency content.

We proposed a detection method for identifying deep network generated images based on their high-frequency characteristics and performed binary classification experiments on datasets of real images and images generated by StyleGAN, StyleGAN2, PGGAN, VQ-VAE2, and ALAE architectures. This detection method achieved an accuracy of 99.2% on uncompressed, high-resolution images with minimal training data, but the accuracy decreased with highly-compressed and/or low-resolution images, although the classifier was able to robustly classify images at resolutions it was not trained at. Finally, we presented a method for modifying the high-frequency spectra of a deep network generated image to mimic the spectra of real images, effectively deceiving the classifier without any visually noticeable changes in the image itself. In the future, this detection and synthesis method will be applied to videos manipulated by deep generative models (i.e. deep fakes) to evaluate their effectiveness.

## Broader Impact

The most apparent impact of the present work is in the application of combating unethical uses of deep network generated images. Since the proposed approach can robustly generalize to unknown generative models and requires minimal training data, it can be easily implemented in browser plugins and mobile applications to warn users that an image is likely fake. This work can be expanded upon for detecting manipulated videos, a trending topic in current research, or towards adversarial training of vision tasks. However, systematic implementation of the proposed method for spoofing images would effectively nullify the capabilities of the classifier, and as a result, could create even more realistic (and harder to detect) deep network generated images for malicious purposes.

Similarly, the current work can be used as a basis for improving the training process of generative models; for example, a metric can be given for generated images in the frequency domain and used for evaluating generative models, and a loss function in Fourier space, weighted towards the highest modes, can be introduced to aid in improving these networks. Generative models for non-image data could benefit from analysis in other domains as well.

A more general result of this work is the conclusion that generative models can have systematic shortcomings that are not immediately evident until observed in other domains (e.g. frequency). In fields where data is scarce and high-quality synthetic data is required to train models, fundamental flaws in synthetic data can have hidden detrimental effects. This opens up questions about the structure of synthetic data and what we perceive to be "high-quality" synthetic data.

## Acknowledgments and Disclosure of Funding

The authors do not acknowledge any outside funding sources or competing interests.

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
