[Reviews · NeurIPS 2020]

Review 1

Summary and Contributions: The paper demonstrates that synthetic images generated by GANs have different decay characteristics in their Fourier spectra. It is able to use this knowledge to build a simple classifier to distinguish between real and fake images, that runs on top of radial decay "shape" parameters from Fourier magnitudes, to find it performs and generalizes well for several current GAN models. It shows that this classifier doesn't work in the presence of compression and other image transformations, and can also be defeated by 'spoofing' the spectrum decay rate. --Post-rebuttal I've read the other reviews and the rebuttal, and retain a positive view of the paper. This is interesting and timely work, and would be of interest to the NeurIPS audience.

Strengths: The paper makes an interesting observation, evaluates it systematically—for regular images, and after compression—uses it to demonstrate that a simple classifier suffices to distinguish real and fake images, but also that it can be easily defeated by 'spectrum synthesis'. This is an insightful and largely well-written paper, that will be of interest to the NeurIPS audience.

Weaknesses: The main weaknesses of this paper are perhaps in missed opportunities to make it even more insightful. - The fact that such a simple classifier works begs the question whether other 'fake image detectors' are fundamentally using the same spectral cues (just in a more opaque way). - This could perhaps be demonstrated by seeing if those detectors also suffered the same degradation in detection accuracy under compression. It would be even more interesting to see if the accuracy of these prior methods suffered after the proposed 'spectrum synthesis' step. - It would also be good to have more discussion, and preferably also analysis, of "why" these generative models produce images with different spectra. Is it something in the small-kernel convolution-transpose decoder architecture that have become so common ? Or is it because of where and how the random source (i.e., noise) is injected into these architectures ? One can imagine experiments to investigate this further: for example, how does fake detection accuracy change when one trains a GAN with spatially larger kernels; and how does it change if one introduces noise in later vs earlier layers.

Correctness: Yes.

Clarity: Yes.

Relation to Prior Work: Yes.

Reproducibility: Yes

Additional Feedback:


Review 2

Summary and Contributions: The paper proposes to use the Fourier transform high-frequency attributes to classify real images and GAN/VAE-generated fake images. Parameters of the decay function fitted on the Fourier spectrum is used in a simple KNN classification. The robustness of the techniques against image transformation such as compression is investigated. They also investigate scaling of the high-frequency components for spoofing attack on the detection.

Strengths: The paper is clear in general. The problem requires attention. Using Fourier frequency for real/fake image detection is reasonable. The synthesis result (Fig. 7) shows the potential of spoofing using scaled high-frequency.

Weaknesses: *Post rebuttal* I have read the rebuttal. Thanks authors for the response. I have slightly increased the score. However, I still have concerns re. the novelty and significance of the results. There have been several works to use frequency analysis (FFT, DCT, DWT) to discriminate real photographic images from various type of fake images (e.g. computer-generated / computer-manipulated images) http://www.ifp.illinois.edu/~ywang11/paper/ICASSP06_Revised.pdf https://link.springer.com/article/10.1007/s11042-019-08354-x http://itiis.org/digital-library/manuscript/file/22012/TIISVol13No2-22.pdf In particular, previous work has shown that frequency analysis can discriminate real/computer-generated images. These images were produced by rendering. Now, with GAN/VAE, we have new ways to use computer to generate images, which quality may not be better than photorealistic rendering. Therefore, given previous research, I have concern re. the novelty and significance of the results. As mentioned, recent work with similar objective has more extensive experiments https://peterwang512.github.io/CNNDetection/ ===================== Novelty is limited. It is rather common to use Fourier transform high frequency component for real/fake classification, e.g. http://www.nws-sa.com/biometrics/facial/SPIE2004.pdf For this work, the difference is just another kind of fake image: images generated by GAN/VAE As this is mostly an experimental study, more models / architectures are needed to achieve reliable conclusion. E.g. this similar work seems to include more models: https://peterwang512.github.io/CNNDetection/ I concern if the technique is sensitive to the decay function fitting method (how do you obtain the fitting parameters?) Results are encouraging, but not great. e.g. only 65.9% for StyleGAN2 even at high resolution

Correctness: yes

Clarity: the paper is clear

Relation to Prior Work: Related work in Fourier transform for real/fake detection needs to be discussed

Reproducibility: Yes

Additional Feedback: Experiments need to be more extensive. Accuracy still has room to improve


Review 3

Summary and Contributions: This paper presents a straightforward observation that images generated by state-of-the-art generative models are quite easy to detect based on their Fourier spectrum falloff. In particular, generated images seem to have much more gentle spectrum decays compared to natural images. The paper shows that a very simple classifier (nearest neighbors based on a few parameters describing the Fourier spectrum radial decay) is quite effective at detecting fake generated images. Comments after rebuttal/discussion: Although there has been substantial work within the forensics community in examining Fourier discrepancies between real and fake images pointed out by other reviewers, I still believe that this observation in the context of GAN generated images is timely and of interest to the NeurIPS community. I think that this paper would definitely be of interest to the community and readers, and I still think that it should be accepted.

Strengths: I think this paper presents a simple and interesting observation about current state-of-the-art generated images that the NeurIPS audience will find interesting. Prior work within the research community seems to have suggested similar findings (observations such as deconvolution checkerboard artifacts in generated images https://distill.pub/2016/deconv-checkerboard/ and high-frequency textures in generated images can be interpreted as "adversarial attacks" on the GAN discriminator), but I think that this paper presents a very piece of evidence that is of interest to the community. I think that one of the strengths of this paper is showing how a very simple approach (fitting a low-parameter decay distribution to the Fourier radial spectrum) provides a simple featurization of images that is surprisingly effective for detecting faked images.

Weaknesses: Although the paper's finding is an interesting observation, it is unclear whether this is just an artifact of current GAN/generative model architectures (and will therefore be resolved with a few more paper iterations), or if there is anything more fundamentally interesting to be analyzed/understood. The difference between the spectral decay of StyleGAN 1 and 2 is quite significant, so it's not unreasonable to think that the next iteration of GAN architectures will make this simple detection technique obsolete. Additionally, as the paper points out, it seems quite easy to break this detection method by just replacing the fake high-frequency content, and the resulting images are still just as convincing to human viewers. I think additional experiments such as showing that this simple detection technique is indeed as effective as more complex learned classifiers on the Fourier spectrum would be of interest to readers.

Correctness: Yes, I think that the claims and experiments are sound and correct.

Clarity: Yes, the paper is generally well-written and easy to understand.

Relation to Prior Work: Yes, although I am not an expert in the area of generative image modeling and GANs, I believe that this paper has done a good job at discussing prior work.

Reproducibility: Yes

Additional Feedback: Line 40: better to include references directly attached to the relevant algorithms instead of putting them all at the end of the sentence. Line 150: This paragraph seems like it is mostly conjecture. I recommend the authors either remove it or attempt to be more precise or provide evidence for this claim. I recommend editing the bibliography for consistency. Some examples are differences in conference venue abbreviations such as [1] vs. [13], and inclusion of page numbers.


Review 4

Summary and Contributions: This paper shows that deep network generated images share an observable, systematic shortcoming in replicating the attributes of these high-frequency modes. Based on this discovery, a detection method relying on the frequency spectrum of the images was proposed, which is able to achieve the high accuracy of detecting whether the image is "fake" or "real".

Strengths: 1. Interesting and reasonable observations. 2. Simple yet effective method for fake image detection. 3. Comprehensive experiments. 4. Discussion on how to improve the fake image generation based on the findings in the paper.

Weaknesses: After reading the rebuttal, may concern is as follows. 1. The observation lacks of detailed explanations. I agree the observation is interesting, but it is pity that there is no theoretical discussion on it. In fact, theoretical discussions will make the paper from an average OK submission to an excellent one. 2. The proposed method is simple. If the paper has an extension, I would like to see the comparison with other trad dictional simple methods in image feature description, e,g,,BoF, LBP, etc.

Correctness: Seems correct.

Clarity: Could be improved on some aspects.

Relation to Prior Work: Adequate. But it is welcomed to introduce more traditional feature extraction methods for image classification for comparison.

Reproducibility: Yes

Additional Feedback: The paper suggests to use compression to improve the generated fake images. But it would be better to use a more explicit way. Also see the weakness.

[Author Response · NeurIPS 2020]

We would like to thank the reviewers for their constructive feedback and comments. In this work, we looked at analyzing
the high-frequency Fourier modes of real and deep network generated images and showed that deep network generated
images fail to replicate the attributes of these high-frequency modes. Using this, we demonstrated that simple classifiers
based on the frequency characteristics of the images can perform effectively and generalize well even when trained with
minimal data. Furthermore, we observed the effects of image transformations on this classification task, showing that
compression has a large impact on the frequency spectra and, consequently, the classifier, whereas the classifier was less
affected by cropping and resolution reduction. To successfully disguise deep network generated images to classifiers
based on the frequency spectra, we proposed a synthesis method for spoofing these images which was able to directly
and more effectively homogenize the spectra in comparison to compression. We are pleased to see that the reviewers
found the observations interesting and insightful to the NeurIPS community (**R1**, **R3**, **R4**), that the problem requires
consideration (**R2**), and the experiment design to be systematic and comprehensive (**R1**, **R4**). Additionally, we are
encouraged to see that the reviewers found the classification method effective even when using simple classifiers (**R1**,
**R3**) and the 'spectrum synthesis' method to show potential in alleviating these spectrum discrepancies (**R1**, **R2**, **R4**).

Some concerns raised by **R1** and **R3** were with regard to the effectiveness of 'more complex' classifiers with respect to
the simple low-dimensional classifier we proposed: whether they are 'using the same spectral cues just in a more opaque
way (**R1**).' **R1** mentioned that this could perhaps be demonstrated by seeing if those detectors also suffered the same
degradation in detection accuracy under compression. For this concern, we refer to the concurrent work of Wang *et al.*
that was briefly referred to in the introduction and by **R2**, where they trained a DNN with a large training set to classify
real and fake images generated by various generative models. Although their work focused on low-resolution images
($224^2$ pixels), their classification accuracy also severely deteriorated with compressed images, but this was partially
alleviated by including compressed images in their training examples. However, we observed similar detrimental effects
on classification at low-resolutions as in their work, and we also showed that compression more severely affected the
spectra of high-resolution images. Therefore, it can be surmised that 'complex classifiers' would suffer even larger
degradation under compression at higher resolution which would indicate that they are in fact using the same spectral
cues in a more opaque way. Hence, we believe it is likely that the proposed 'spectrum synthesis' method would also
detrimentally affect their accuracy. In further work, we could directly compare those classification methods.

In relation to the aforementioned Wang *et al.* paper, **R2** raised concerns that more generative models are needed to
achieve a more reliable conclusion. However, to systematically analyze the effects of higher-resolution and compression
on images, we required datasets or pretrained models that could provide *uncompressed, high-resolution* images, which
limited the number of models we could evaluate relative to Wang *et al.* whose work was focused towards low-resolution
images. To our knowledge at the time, of the six models which were capable of generating high-resolution ($1024^2$)
images, we considered five for the manuscript. For the sixth model (IntroVAE), the images were only available in a
heavily compressed form without an available pretrained model and hence were not suitable for our investigation.

Furthermore, we would like to clarify some of the results. **R3** mentioned the difference in *decay* between StyleGAN
and StyleGAN2, indicating that next generation GAN architectures would solve this discrepancy problem. However,
we note that the differences (in Fig. 5a) between these two models are actually along the $b_1$ axis, which indicates the
level of high-frequency content. The decay rate $b_2$, the primary discrepancy between real and fake images, is largely
unchanged between the two models. Therefore, we don't believe that there is indication that this problem is being
implicitly solved in the current development path of these models. **R2** noted the poor performance (65.9%) of the
classifier on highly-compressed StyleGAN2 images at high-resolutions. We reiterate that this is a level of compression
where visually noticeable artifacts are observed, and it is not feasible to use it in cases where high-quality synthetic
data is needed. In the literature on fake image detection, classification is usually performed using the original output
of the generative models, for which we obtain a 97.4-99.9% classification accuracy. **R2** also raised the question of
the sensitivity of the decay function fitting method. For this, we've looked at various decay functions (power law,
exponential, etc.) and classifier and fitting parameters. As long as the fitting window (threshold frequency) was
sufficiently high such that it was primarily fitting noise, the results showed very little sensitivity (on the order of $\pm 1\%$).

There was interest from **R1**, **R3**, and **R4** in having further discussion on explaining the behavior of the generative
models shown in the paper. We posit that it is related to the checkerboard artifacts shown by Odena *et al.* as mentioned
by **R3** and the AutoGAN method by Zhang *et al.*, which can be seen as an inverse of our 'spectrum synthesis' method.
Additionally, we will note the formatting advice of **R3** and the request for more related work in Fourier space image
forensics from **R2**. Although **R2** has pointed out that using the Fourier space for image forensics has been done before,
this work performs it in relation to deep network generated images for which the mechanisms and behavior of the high-
frequency content is quite different than in biometric spoofing attacks. **R3** also pointed out the classification can easily
be defeated using the proposed spoofing method, but we believe that is a positive aspect of the work as it shows that
these generative models need some post-processing of their original images. **R4** mentioned the need for a more explicit
way to improve the spectra of fake images instead of compression, but we believe that our 'spectrum synthesis' method
addresses this. We acknowledge the feedback from the reviewers and hope that these points address their concerns.

[Meta-Review · NeurIPS 2020]

The reviewers are in agreement that the observations made in this paper (fake images generated by GANs and VAEs are detectable from their high frequency statistics) are interesting and relevant to the NeurIPS community. After significant reviewer discussion, and after weighting the author response, the reviewer consensus is that the paper should be accepted. The reviewers did not identify any major shortcomings of the paper that were left unaddressed. I agree that the paper presents interesting results and I support acceptance.